# The translation and validation of the Organ Transplant Symptom and Well-Being Instrument in China

Ying Shi[1], Zhang Dan[2], Zijun Tao[2], Qi Miao[2], Tiantian Chang[2], Xu Zhang[2], Xiaoyu Jiang[2], Xiaofei Li[1]*

1 Department of Intensive Care Unit, Chongqing University Cancer Hospital Hospital, Chongqing, China,
2 Department of Transplantation and Hepatobiliary, The First Affiliated Hospital of China Medical University, Shenyang, Liaoning, P. R. China

* lixiaofei0603@aliyun.com

## Abstract

To translate the Organ Transplant Symptom and Well-Being instrument (OTSWI) into Chinese and test the reliability and validity of the Chinese version. A total of 259 patients with organ transplants were recruited from The First Affiliated Hospital of China Medical University in Shenyang, from November 2020 to January 2021. Construct validity was evaluated using exploratory factor analysis (EFA) and reliability were assessed using test-retest reliability and internal consistency. The Cronbach's α of the Chinese version of the Organ Transplant Symptom and Well-being instrument was 0.93. EFA demonstrated that 80.785% of the total variance was explained by a seven-factor solution. The criterion validity of the SF-36 was -0.460 ($p < .01$), while the test-retest reliability was 0.710. The Chinese version of the OTSWI questionnaire is a valid and reliable instrument for assessing the quality of life of organ transplant patients for symptoms and well-being in China.

**Data Availability Statement:** Our data contains sensitive patient information, which was collected through the Ethics Committee of the First Affiliated Hospital of China Medical University. Please

## Introduction

Organ transplantation has emerged as an acceptable treatment option for patients with end-stage disease processes, such as end-stage renal disease (ESRD), acute and chronic decompensated liver disease, end-stage cardiac and pulmonary diseases, and diseases of the pancreas and intestines [1, 2]. According to a survey from the Global Observatory on Donation and Transplantation (GODT), 163,141 organ transplants were performed globally in 2019 [3]. Although the number of transplants decreased due to the impact of COVID-19 in 2020, the number of transplants tend to increase annually, in general. The goal of transplant surgery is to maximize the length and quality of life of transplant recipients. The quality of life of organ transplant patients following transplantation is an important research topic in the medical field and should be attended to.

Health- related quality of life (HRQOL) is defined as the individual's subjective evaluation of the impact of his/her physical, psychological, social, and environmental areas of functioning and well-being [4]. HRQOL focuses on the impact of health conditions and their symptoms on a patients' well-being [5]. HRQOL in organ transplant recipients has been extensively

contact the Ethics Committee of the article with any questions. The contact point for the data required by the editor is the Ethics Committee of the First Affiliated Hospital of China Medical University. The contact number is 024-83282837, mailing of ethics committee mailing address: Shenyang City, Liaoning Province, peace zone, Nanjing North Street 155, zip code: 110001.

**Funding:** The authors received no specific funding for this work.

**Competing interests:** The authors have declared that no competing interests exist.

assessed since it emergence as an important outcome for measuring the long-term success of transplants [6–10]. Previous research on the quality of life as it relates to organ transplantation primarily includes the quality of life of the long-term survival of patients following transplantation [10–16], the effect of compliance with immunosuppressive drugs as it relates to the quality of life [17–21], and other influencing factors of quality of life in patients following transplantation [4, 22–24]. The most common tool for measuring the quality of life of patients following an organ transplant is the Short Form 36 (SF-36) [25–29], however, it is less sensitive to disease-specific HRQOL issues. With the rapid development of transplantation technology, the survival rate of transplant patients has significantly improved and relevant scholars have begun to develop domain-specific scales for organ transplantation, such as the Post-Liver Transplant Quality of Life (PLTQ) [30, 31] and the End-Stage Renal Disease Symptom Checklist-Transplantation Module (ESRD-SCL) [32–34]. Saab et al. [9] developed the PLTQ in 2011, which includes eight dimensions on 32 items, to measure the quality of life of liver transplant recipients. Peng [35] finished the Chinese version of the scale in 2014. Ya-bin et al. [30] used this scale to examine the quality of life of liver transplant patients and found a need to improve the quality of life of patients following a liver transplant, with economic issues cited as the primary concern of patients after liver transplantation. Franke et al. [32] developed the ESRD-SCL in 1999, which measures the quality of life in kidney transplant recipients. In 2006, Xu et al. [36] adjusted the original scale and compiled a scale suitable for a Chinese cultural background. This new scale evaluates the quality of life of patients before and after kidney transplantation. However, the types of organ transplants not only include livers and kidneys, but also hearts, lungs, and other organs, and the quality of life of these patients need to be measured as well. Unfortunately, there is currently no Chinese version of a scale that measures the quality of life for all transplant types.

The Organ Transplant Symptom and Well-Being Instrument (OTSWI), which is an instrument that combines the measurements of symptom distress and well-being as it pertains to the health-related quality of life was developed by Anna Forsberg [37]. A symptom is defined as a physical or mental feature that is regarded as indicating a condition or disease, particularly when such a feature is apparent to the patient [38]. Symptoms maybe defined as an experience reflecting changes in the biopsychosocial functioning, sensations, or cognition of an individual. Symptoms are a regular part of the human experience [37]. Symptom experience involves two concepts, symptom occurrence and symptom distress. Symptom occurrence (the cognitive pathway of symptom experience) is described as the frequency, severity, and duration of a given symptom as perceived by an individual. Symptom distress (the emotional pathway of symptom experience) demonstrates how recipients are influenced daily by their symptoms [39]. Neurological symptoms are the most common complications seen in organ transplant patients [40]. Neurologic symptoms can include neurotoxicity of immunosuppressive agents, seizures, encephalopathy, cerebrovascular events, opportunistic infections, post-transplant lymphoproliferative disorder, and central pontine myelinolysis [41]. Most of the tools used to assess symptom experiences are based on clinical experience and experts and tend to only focus on the symptoms [31, 42]. The OTSWI has been demonstrated to be an effective tool for measuring symptom distress and well-being. Forsberg et al. [43, 44] used the OTSWI and the Pain-O-Meter to suggest that lung recipients with pain reported lower well-being and higher symptom distress, but were not more fatigued than those without pain.

An extensive review demonstrated that only a few studies have investigated the symptom experience of organ transplant recipients [12, 45–47]. Therefore, it is important to measure symptomatic distress and well-being in combination with the health-related quality of life in organ transplant patients. The OTSWI has been validated in Sweden, with Cronbach's alpha ranging from 0.81 to 0.92. In the original scale, the authors performed a factor analysis and

extracted eight factors, but did not perform a validated factor analysis. Therefore, we hypothe-sized that the currently published factor structure may not be appropriate for the Chinese organ transplant patient population, and we performed statistical analyses to determine whether previous models of OTSWI are appropriate for data from the Chinese organ trans-plant patient population. In order to find a suitable tool to evaluate the symptomatic distress and well-being of Chinese organ transplant patient population.

In order to use the OTSWI to measure the symptoms and well-being of the quality of life of transplant patients among Chinese patients, the original version was translated to a Chinese version. The current study consists of two parts: 1) to translate the English version of the OTSWI into Chinese and make cultural adjustments; and 2) to evaluate the validity and reli-ability of the Chinese version of the OTSWI (C-OTSWI) for use with Chinese patients.

## Methods

### Aim

To translate the Organ Transplant Symptom and Well-Being instrument (OTSWI) into Chi-nese and test the reliability and validity of the Chinese version.

### Methodology

**Study design.**  This study used a descriptive, cross-sectional design to study validity and reliability of the newly translated OTWSI. From November 2020 to January 2021, some recipi-ents who fulfilled the inclusion criteria of this study at the First Affiliated Hospital of China Medical University were invited to participate in this survey.

**Translation process.**  The English version of the OTSWI was translated into Chinese with permission from Anna Forsberg, who developed the original OTSWI. The translation process of the OTSWI was conducted by first doing a forward translation, then a back translation, then an expert committee assessment, and finally pretesting [48].

First, the forward translation of the original version of the OTWSI into Chinese was com-pleted by two independent translators, including a certified translator (linguist) and a clinician (for any medical terms), both of whom are bilingual, bicultural, native Chinese speakers. The translators were informed of the study objectives, received the original version of the OTSWI via e-mail, and worked independently from one another. After the translations, all members of the research team reviewed and discussed any incongruity in the two copies until a consensus was reached.

The translated version was then back-translated, blindly, into English by two experts in the medical field. One studied and worked in an English-speaking country for 15 years, while the other has been working in a clinical setting for 30 years in China. The two back-translation versions were then compared, verified, and revised by the research team and a back-translated version was finally sent to Dr. Forsberg to confirm.

An expert panel, composed of the translators and the researchers, then met to compare the Chinese translated versions. The best translations were merged, after which a discussion about the cultural adaptation of several terms was carried out until a consensus was reached. At the end of the meeting, a final harmonized Chinese version of the OTSWI was obtained. Item 14 ("Due to my physical condition I can't take a bath or shower") of the original questionnaire was adapted to "Due to my physical condition I can't take a shower" to better fit the Chinese cultural environment.

Finally, ten patients took the pilot of the Chinese version of the OTSWI in order to discuss the appropriateness of the items and to identify comprehension problems for organ transplant patients. These patients were selected randomly from the target population. Each patient was

asked whether the words and terms used in the C-OTSWI were clear, relevant, and understandable. In this way, patients identified their difficulties and discussed them afterwards. All ten subjects agreed that the C-OTSWI was straightforward and easy to understand. The average time to complete the questionnaire was 15 minutes.

The C-OTSWI was then tested for reliability and validity among 259 subjects in the same institute, all of whom were selected from the patients referred to the Organ Transplantation Department of The First Affiliated Hospital of China Medical University.

## Participants and sample size

The current study was conducted in a single center. Our participants were all from The First Affiliated Hospital of China Medical University in Shenyang. The data were collected from November 2020 and January 2021.

The required sample size for the study was estimated by the rule of at least a ratio of five participants per measured item, therefore more than 200 patients were planned to be included. Patients were asked to fill out the patient information sheet as well as the C-OTSWI, which took them approximately 15–20 minutes to complete. All of subjects were selected from the patients referred to the Organ Transplantation Department of The First Affiliated Hospital of China Medical University during November 2020 to January 2021.

Eligibility inclusion criteria included the following: (1) age $\geq$ 18 years old; (2) undergoing their first organ transplantation; (3) successful functioning of the transplanted organ(s); (4) normal reading and writing ability, good language communication, and able to complete the questionnaire independently or under the guidance of researchers; (5) $\geq$ 1 month after surgery; and (6) completed the informed consent forms. Exclusion criteria included the following: (1) a history of neurological or psychiatric disorders; and (2) currently has an acute illness.

## Instrument

**Organ Transplant Symptom (OTWSI).**   The OTSWI includes nine dimensions with 40 items: fatigue (3 items), joint and muscle pain (3 items), basic activities in daily life (3 items), sleeping problems (3 items), cognitive functioning (2 items), mood (2 items), foot pain (2 items), economics (2 items), and symptoms (20 items). For each question, answers are ranked using a 5-point Likert scale, with scores of 0 (not at all), 1 (a little), 2 (somewhat), 3 (quite a bit), and 4 (very much). A higher total score indicates a poorer quality of life.

**Short Form 36 health survey questionnaire (SF-36).**   SF-36 include 36 questions and categorized into eight-domain: physical functioning (PF,10 items), general health (GH,5 items), role physical (RP; 4 items), bodily pain (BP, 2 items), social functioning (SF,2 items), vitality (VT,4 items), role emotional (RE,3 items), and mental health (MH,5 items). The scale has good reliability and validity in the Chinese population, with consistency reliability coefficients ranging from 0.72 to 0.88 for all dimensions except social functioning and living as dimensions. The higher score indicating better health [49].

**Ethical considerations.**   A total of 259 organ transplant patients participated in the current study. The participants were recruited from The First Affiliated Hospital of China Medical University. The study was approved by the Ethics Committee of The First Affiliated Hospital of China Medical University (202157). Prior to enrollment, participants were informed of the study objectives, anonymity of data collection management, and their free right to withdraw at any time from the participation. All participants voluntarily signed the informed consent form.

**Statistical methods.**   Data were analyzed using SPSS version 25.0. Demographic characteristics and clinical data are presented using means and standard deviations for continuous

variables and percentages or frequencies for categorical variables. The item analysis consisted of item-total correlation and homogeneity, while test-retest reliability was conducted and internal consistency was assessed using Cronbach's alpha coefficient. To analyze the C-OTSWI factor structure, a principal components factor analysis was conducted using a varimax rotation. Sampling adequacy was measured by calculating a Kaiser-Meyer-Olkin (KMO) index [50]. Statistical significance was set at $p < 0.05$. The criterion validity was tested using the Short Form 36 (SF-36) and total scale scores for the C-OTSWI and the SF-36 were both calculated according to standard scoring procedures.

**Reliability.** In the current study, internal consistency was assessed by Cronbach's α coefficient to indicate the degree of correlation between the items. The item-total correlation is a measure of liability of a multi-item scale. In the current study, the item-total correlations were used to assess the item internal consistency, which measures the correlation between the score of an individual item and the score of its own scale, corrected for overlap (the sum of the scores of the remaining items that form the scale). A value of $> 0.30$ was considered to be substantial [51, 52].

**Construct validity.** Factor analysis assesses constructing validity for a questionnaire, which tests the hypothesis that there is a relationship between the observed variables and their underlying constructs [52]. In the current study, exploratory factor analysis (EFA) was conducted to test the structure of the Chinese-version of the OTSWI. A factor was considered being significant and retained only if it had an eigenvalue $> 1$ and a variable with factor loading $> 0.50$ on either a positive or a negative factor [53]. The criterion-related validity between the C-OTSWI and SF-36 was assessed using spearman correlation analyses.

## Results

### Item analysis

The questions were analyzed in terms of the correlations among item scores, the total instrument score, and homogeneity testing (Table 1). A correlation coefficient of lower than 0.3 was adopted as the basis for question deletion. All of the item scores were higher than 0.3, except for items 26 and 27. Clinically, organ transplant patients tend to have oral problems after surgery and their appetite may increase after taking hormones, which are part of the symptoms. Therefore, the conclusion was made that no items should be deleted.

Sample consisted of 259 patients. Within this sample, 35% were women, most patients (42%) were aged 18–45 years, 77% were married, the largest education group had an undergraduate level education (accounting for 35% of the sample), the employment rate was 40%, and 92% lived with family (Table 2).

### Internal consistency

Cronbach's α value was 0.934 for the entire sample and ranged from 0.726–0.861 for the factors measuring sleep, joint muscle pain, foot pain, fatigue, cognition, daily basic activities, mood, economy, and symptoms.

### Test-retest reliability

In all, 30 participants completed the retest questionnaire in order to assess the test-retest reliability over a two-week interval. A Spearman's rank correlation coefficient (r) of 0.30 or lower is considered weak, between 0.30 and 0.60 is considered moderate, and $\geq 0.60$ is considered strong [54]. The intraclass correlation coefficients showed a high test-retest reliability ($r = 0.713$, $p < .001$) for the C-OTSWI.

**Table 1. Organ Transplant Symptom and well-being instrument: Item analysis.**

| Items | Items mean | SD | Corrected item-total correlation | Cronbach's if item deleted |
|---|---|---|---|---|
| **Total scale (Cronbach's α = 0.952) Sleeping problems (Cronbach's α = 0.802)** | | | | |
| 1.I have difficulties with falling asleep | 0.776 | 1.051 | 0.515 | 0.932 |
| 2.I sleep poorly | 0.961 | 1.137 | 0.574 | 0.931 |
| 3.I wake up during the night | 1.494 | 1.205 | 0.427 | 0.933 |
| **Ache of lower limb (Cronbach's α = 0.841)** | | | | |
| 4.My muscles are aching | 0.641 | 0.968 | 0.565 | 0.932 |
| 5.My joints are aching | 0.792 | 1.032 | 0.583 | 0.931 |
| 6.My legs are aching | 0.703 | 1.016 | 0.553 | 0.932 |
| 7.There is a numb and stabbing feeling in my feet | 0.421 | 0.770 | 0.484 | 0.932 |
| 8.There is a burning ache in my feet | 0.529 | 0.877 | 0.446 | 0.933 |
| **Fatigue (Cronbach's α = 0.940)** | | | | |
| 9.I am physically tired | 1.174 | 1.163 | 0.717 | 0.930 |
| 10.I have no energy | 0.988 | 1.094 | 0.720 | 0.930 |
| 11.I feel lazy and listless | 0.934 | 1.071 | 0.702 | 0.930 |
| **Cognitive functioning (Cronbach's α = 0.757)** | | | | |
| 12.I have difficulties to remember | 1.602 | 1.267 | 0.579 | 0.931 |
| 13.I find it hard to concentrate | 0.946 | 1.077 | 0.612 | 0.931 |
| **Basic activities in daily life (Cronbach's α = 0.824)** | | | | |
| 14.Due to my physical condition I can't take a shower | 0.232 | 0.716 | 0.317 | 0.934 |
| 15.Due to my physical condition I can't get dressed by myself | 0.097 | 0.443 | 0.375 | 0.933 |
| 16.Due to my physical condition I can't buy food by myself | 0.097 | 0.683 | 0.382 | 0.933 |
| **Mood (Cronbach's α = 0.861)** | | | | |
| 17.I feel irritated | 0.919 | 1.063 | 0.593 | 0.931 |
| 18.I feel angry | 0.498 | 0.978 | 0.420 | 0.933 |
| **Economy (Cronbach's α = 0.727)** | | | | |
| 19.I worry about not being able to keep my job due to my health condition | 1.077 | 1.339 | 0.505 | 0.932 |
| 20.I worry about my economy due to my health condition | 0.745 | 0.963 | 0.524 | 0.932 |
| **Symptom (Cronbach's α = 0.861)** | | | | |
| 21.I'm breathless | 0.502 | 0.502 | 0.617 | 0.931 |
| 22.I need to rest because I am breathless | 0.394 | 0.777 | 0.621 | 0.931 |
| 23.I'm swollen | 0.510 | 0.891 | 0.340 | 0.933 |
| 24.I feel nauseous | 0.382 | 0.755 | 0.542 | 0.932 |
| 25.I have oral fungus | 0.425 | 0.847 | 0.393 | 0.933 |
| 26.I have oral herpes | 0.394 | 0.792 | 0.258 | 0.934 |
| 27.I have increased appetite for food | 1.541 | 1.376 | 0.203 | 0.936 |
| 28.I have decreased appetite for food | 0.440 | 0.844 | 0.459 | 0.933 |
| 29.I have dyspepsia | 0.807 | 1.039 | 0.516 | 0.932 |
| 30.I'm constipated | 0.637 | 0.992 | 0.329 | 0.934 |
| 31.I have diarrhea | 0.811 | 1.117 | 0.414 | 0.933 |
| 32.My skin is itching | 0.965 | 1.125 | 0.411 | 0.933 |
| 33.I have headache | 0.668 | 0.922 | 0.522 | 0.932 |
| 34.There is a burning pain in my hands | 0.313 | 0.652 | 0.454 | 0.933 |
| 35.There is a numb and stabbing pain in my hands | 0.471 | 0.886 | 0.446 | 0.933 |
| 36.My hands are trembling | 1.073 | 1.260 | 0.382 | 0.934 |
| 37.I feel dizzy | 0.525 | 0.850 | 0.639 | 0.931 |
| 38.I feel sad | 0.556 | 0.956 | 0.698 | 0.930 |
| 39.My looks makes me embarrassed | 0.525 | 0.978 | 0.580 | 0.931 |
| 40.My libido is decreased | 1.151 | 1.293 | 0.594 | 0.931 |

**Table 2. Demographic and clinical characteristics of participants.**

| Characteristics | N | % |
|---|---|---|
| **Gender** | | |
| male | 168 | 64.9% |
| female | 91 | 35.1% |
| **Age(years)** | | |
| 18–45 | 108 | 41.7% |
| 45–55 | 89 | 34.4% |
| > = 55 | 62 | 23.9% |
| **Education** | | |
| Junior middle school and below | 40 | 15.4% |
| Senior middle school | 51 | 19.7% |
| College/advanced degree | 168 | 64.9% |
| **Marital status** | | |
| single | 41 | 15.8% |
| married | 199 | 76.8% |
| divorce | 17 | 6.6% |
| widowed | 2 | 0.8% |
| **The state of work** | | |
| employee | 103 | 39.8% |
| unemployed | 57 | 22% |
| retirement | 92 | 35.5% |
| farming | 7 | 2.7% |
| **Way of living** | | |
| living with your family | 237 | 91.5% |
| share with others | 1 | 0.4% |
| living alone | 21 | 8.1% |
| **Hobby** | | |
| smoke | 15 | 5.8% |
| drink alcohol | 7 | 2.7% |
| smoke and drink alcohol | 3 | 1.2% |
| neither smoke nor drink alcohol | 234 | 90.3% |
| **Exercise** | | |
| Regular (≥3-4per/week, every time≥20min) | 130 | 50.2% |
| Occasional (≥1-2per/week, every time≥20min) | 98 | 37.8% |
| never | 31 | 12.0% |
| **Type of Organ Transplantation** | | |
| liver transplant | 111 | 42.9% |
| kidney transplant | 136 | 52.5% |
| heart transplant | 3 | 1.2% |
| lung transplant | 5 | 1.9% |
| Co-transplantation of pancreas and kidneys | 4 | 1.5% |

## Factor analysis

Factor loadings were extracted from the polychromic correlation matrix of the items by using a principal component analysis with a varimax rotation. As shown in Table 3, an eight-factor solution best fit the 40 items, accounting for 80.79% of the total variance of the items, which is satisfactory. In the original scale, the author divided the eight dimensions of items 1–20, while

**Table 3. Principle components analysis followed by varimax rotation factor loadings of Chinese organ transplant symptom and well-being instrument.**

| Rehabilitation Tool | F1 | F2 | F3 | F4 | F5 | F6 | F7 | F8 |
|---|---|---|---|---|---|---|---|---|
| 1.I have difficulties with falling asleep | | | | 0.896 | | | | |
| 2.I sleep poorly | | | | 0.826 | | | | |
| 3.I wake up during the night | | | | | | | | 0.809 |
| 4.My muscles are aching | 0.712 | | | | | | | |
| 5.My joints are aching | 0.694 | | | | | | | |
| 6.My legs are aching | 0.794 | | | | | | | |
| 7.There is a numb and stabbing feeling in my feet | 0.755 | | | | | | | |
| 8.There is a burning ache in my feet | 0.685 | | | | | | | |
| 9.I am physically tired | | 0.810 | | | | | | |
| 10.I have no energy | | 0.844 | | | | | | |
| 11.I feel lazy and listless | | 0.870 | | | | | | |
| 12.I have difficulties to remember | | | | | | | 0.765 | |
| 13.I find it hard to concentrate | | | | | | | 0.651 | |
| 14.Due to my physical condition I can't take a shower | | | 0.831 | | | | | |
| 15.Due to my physical condition I can't get dressed by myself | | | 0.832 | | | | | |
| 16.Due to my physical condition I can't buy food by myself | | | 0.889 | | | | | |
| 17.I feel irritated | | | | | 0.839 | | | |
| 18.I feel angry | | | | | 0.871 | | | |
| 19.I worry about not being able to keep my job due to my health condition | | | | | | 0.871 | | |
| 20.I worry about my economy due to my health condition | | | | | | 0.839 | | |
| Eigenvalue | 7.217 | 2.112 | 1.679 | 1.494 | 1.115 | 1.033 | 0.777 | 0.688 |
| Percentiles of variance. % | 36.085 | 10.561 | 8.394 | 7.470 | 5.782 | 5.166 | 3.883 | 3.442 |
| Total variance, % | | | | | | | | 80.785 |

items 21–40 are grouped into the symptoms, acting as single items. The dimensions of sleeping problems, fatigue, cognitive function, basic daily activities, economics, and emotion were consistent with the original scale. The sleep problems dimension included items 1–3, but in our EFA, item 3 was not assigned to the sleep problems dimension. In the original scale, the dimension of joint and muscle pain included 3 items (4–6), but in our factor analysis items 7 and 8 were also included as joint and muscle pain. Therefore, the dimension of joint and muscle pain included five items, which are renamed as lower extremity joint muscle pain for the C-OTSWI. The remaining 20 items were classified as symptom

## Discussion

The OTSWI is an instrument that combines the symptom experience and well-being measurements on the concept of HRQOL [37]. HRQOL is an important construct in measuring the success of transplantation in the long term. Vermeulen et al. [55] argue that little evidence is provided on the impact of symptom experience on outcomes in terms of HRQOL. Chen et al. have demonstrated that one of the reasons for the low HRQOL is the experience of complicated symptoms [45]. However, only a few studies to date have investigated the symptom experience of organ transplantation recipients. Typically, the nurse will conduct a comprehensive assessment of the symptoms experienced from the perspective of the patient and then will determine the intervention strategy based on their assessment results. The OTSWI provides a useful tool to assess post-transplant patients and to aid transplant nurses in planning their goal-directed nursing interventions in collaboration with the patients. Therefore, it is necessary to translate the OTSWI into Chinese for use in China.

The current study translated the English version of the OTSWI into Chinese and validated the Chinese version. The C-OTSWI exhibited a high degree of internal consistency reliability and the test-retest reliability. In the current study, the questionnaire was slightly modified to ensure its suitability for a Chinese cultural context, including changing item 14 ("Due to my physical condition I can't take a bath or shower") to "Due to my physical condition I can't take a shower". Pre-test interviews were conducted to record patients' facial expressions and any misunderstandings of the scale. Through this procedure, the content of the questionnaire was determined to be applicable to native Chinese speakers.

Cronbach's alpha, item-total correlations, and test-retest procedures were used to assess the scale reliability. The internal consistency coefficient (Cronbach's α) of the original version of the OTSWI was 0.81–0.92, and Cronbach's α of the C-OTSWI was 0.86–0.94, suggesting that the Chinese version of the OTSWI has good reliability in this population. The results of the test-retest (r = 0.713, $p < .001$) demonstrated that the stability of responses to the items was good.

An EFA was conducted to identify potential sub-scales of the OTSWI. Findings from the current study showed that the data didn't fit to the original nine-factor model. One explanation for this difference could be due to cultural differences. Through exploratory factor analysis of the C-OTSWI, 40 items of the scale were divided into eight factors. The results of the EFA demonstrated that the C-OTSWI has acceptable structural validity. In the original scale, the author found that items 1–20 can be divided into eight dimensions, but in the Chinese version of the scale, the joint muscle pain and foot pain belonged to the same dimension. Some experts, during consultation, asked about the joint muscle pain and foot pain in the original scale, all of which primarily inquired about the lower extremity pain. We considered that these items all related to the description of lower extremity pain, so those items were grouped into the same dimension and renamed it "lower extremity joint muscle pain". In addition, item 3 ("Waking up in the middle of the night") was not included in the sleep problems dimension. Sleep is a state of consciousness and has a profound impact on an individual's health, well-being, and quality of life. Sufficient duration of and good quality of sleep is a well-recognized predictor of physical and mental health [56]. However, insomnia, as the most common sleep disorder, is a public health problem [57]. The American Academy of Sleep has stated that the presence of a long sleep latency, frequent nocturnal awakenings, or prolonged periods of wakefulness during the sleep period, or even frequent transient arousals are evidence of insomnia [57]. Therefore, we included "Waking up in the middle of the night" as a sleep problems dimension. The remaining 20 items were named symptoms in the original scale, while they were used to measure symptom distress in the Chinese version. Therefore, the Chinese version of the scale includes eight dimensions with a total of 40 items. A principal component analysis and varimax rotation method were used to evaluate the factor loadings of each item; each item loading was higher than 0.35. There was a negative and significant correlation between the C-OTSWI and SF-36.

## Limitations

Some limitations of the current study should be acknowledged. Data from post-transplant patients were only collected from one hospital in the northeastern region. It is a single-center study which may have delivered biased; a multicenter random sampling study should be adopted in future studies. Second, a cross-sectional design was used in the current study. In future studies, longitudinal studies should be conducted. Thirdly, the test showed that the Chinese version of the scale was usable, and although the correlation coefficient between the entries and the overall was low in the scale, such problems were found in the communication

with the patients, so the entries were retained. In future studies, we can conduct a multicenter survey to test this further.

## Conclusions

The Chinese version of the OTSWI is adequately translated, reliable, and valid. We are confident that this Chinese version of the OTSWI is suitable to test the quality of organ transplant patients.

## Supporting information

**S1 Text. Chinese version of the OTSWI.**
(DOCX)

**S2 Text. English version of the OTSWI.**
(DOCX)

## Acknowledgments

A heartfelt thanks are given to all of the recipients who participated in the current study and all the teachers from The First Affiliated Hospital of China Medical University who helped with this study. Without them, this study would not have been possible.

## Author Contributions

**Conceptualization:** Ying Shi.

**Data curation:** Qi Miao, Tiantian Chang.

**Investigation:** Zhang Dan, Zijun Tao, Xu Zhang, Xiaoyu Jiang.

**Supervision:** Xiaofei Li.

**Validation:** Xiaofei Li.

**Writing – original draft:** Ying Shi.

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
