## [Decision Letter · Decision Letter 0]

25 Apr 2022

PGPH-D-21-00455

The translation and validation of the Organ Transplant Symptom and Well-Being instrument in China

Dear Dr. LI,

Thank you for submitting your manuscript to PLOS Global Public Health. After careful consideration, we feel that it has merit but does not fully meet PLOS Global Public Health’s publication criteria as it currently stands. Therefore, we invite you to submit a revised version of the manuscript that addresses the points raised during the review process.

We look forward to receiving your revised manuscript.

Kind regards,

Javier H Eslava-Schmalbach, M.D., Ph.D., MSc

Academic Editor

Journal Requirements:

1. Please send a completed 'Competing Interests' statement, including any COIs declared by your co-authors. If you have no competing interests to declare, please state "The authors have declared that no competing interests exist".

3. We do not publish any copyright or trademark symbols that usually accompany proprietary names, eg (R), (C), or TM  (e.g. next to drug or reagent names). Please remove all instances of trademark/copyright symbols throughout the text, including (C) on page 21.

4. In the online submission form you indicate that your data is not available for proprietary reasons and have provided a contact point for accessing this data. Please note that your current contact point is a co-author on this manuscript. According to our Data Policy, the contact point must not be an author on the manuscript and must be a third party. Please revise your data statement to a non-author institutional point of contact, such as a data access or ethics committee, and send this to us via return email. Please also include contact information for the third party organization, and please include the full citation of where the data can be found.

Additional Editor Comments (if provided):

Dear author:

We have finally received comments from our reviewers: Please respond/include each one of their observations. Also, please respond/include these suggestions:

1. Check the correct use of English in the manuscript, so the adequate use of terms like: farm, retire, man, woman

2. The test-retest reliability section should be corrected. You suggest to use one test, and finally ended up using other.

3. References: 3, 13, 53, should be double-checked

4. Include texts in supplementary material, in both languages

Reviewers' comments:

Reviewer's Responses to Questions

**Comments to the Author**

1. Does this manuscript meet PLOS Global Public Health’s publication criteria? Is the manuscript technically sound, and do the data support the conclusions? The manuscript must describe methodologically and ethically rigorous research with conclusions that are appropriately drawn based on the data presented.

Reviewer #1: Yes

Reviewer #2: Yes

2. Has the statistical analysis been performed appropriately and rigorously?

Reviewer #1: No

Reviewer #2: Yes

3. Have the authors made all data underlying the findings in their manuscript fully available (please refer to the Data Availability Statement at the start of the manuscript PDF file)?

Reviewer #1: Yes

Reviewer #2: Yes

4. Is the manuscript presented in an intelligible fashion and written in standard English?

Reviewer #1: Yes

Reviewer #2: Yes

5. Review Comments to the Author

Reviewer #1: All comments and suggestions are in the attached files. The quality of the study was evaluated applying the STROBE checklist and the recommendations by section were abandoned.

1. It is considered important that the authors include a paragraph on the control of bias in the study.

2. The study proposes a factorial analysis. Given that there is a hypothesis about the factorial structure of the instrument, it is suggested being explicit about that hypothesis and indicate that it is NOT an exploratory analysis, but a confirmatory one; and make explicit why it was decided to do a factorial analysis of principal components using a varimax rotation.

Finally, ¿why does the study use the Pearson correlation and not the Spearman correlation if the variables are ordinal? I suggest change that correlation or show that the variables meet the assumptions.

3. In the methodology the authors say that “A value of > 0.40 was considered to be substantial” and then in the results the authors say that “A correlation coefficient of lower than 0.3 was adopted as the basis for question deletion”

Why they change the first assumption? A correlation of 0.3 is considered low, and a candidate for deletion.

The authors say “Correlations between individual items and the total score of the C-OTSWI were significant and positive for the entire sample.”

But It’s not the same the statistical significance and the strength of association. Seven items have a correlation between individual items lower that 0.4 and two lower that 0.3; It’s important analyse is it’s better eliminate those items or explain very well why they don’t do that.

In the methodology the authors say that they will use a Pearson correlation, but in the results, they write about a Spearman’s rank correlation coefficient and then report an intraclass correlation coefficients as a result. They must change that and be consistent between methodology and results.

4. The authors include some limitations, but nothing about the lack of bias control

5. The authors say: “The high correlation coefficient shows that the items of the scale have a strong relationship to the scale construct.”

There was not high correlation in all cases, my recommendation is to be conservative in the discussion and more precise based on the results

Reviewer #2: The manuscript responds to a need in evaluating the outcomes and prognosis of transplant activity. The use of the OTSWI instrument has broadened the horizon of the evaluation of transplant patients beyond physical well-being and allows the senses of physical, mental and emotional well-being to be included. The methodology used both to adapt the scale from the point of view of the language and with the linguistic and cultural aspects of the conception of the disease, is absolutely appropriate. Regarding the methodology to evaluate the validity and internal consistency, reliability and concurrent validity, the approach, the results and the significance are interesting and clearly important and applicable.

6. PLOS authors have the option to publish the peer review history of their article (what does this mean?). If published, this will include your full peer review and any attached files.

**Do you want your identity to be public for this peer review?** For information about this choice, including consent withdrawal, please see our Privacy Policy.

Reviewer #1: No

Reviewer #2: No

---

## [Editor Report · Decision Letter 1]

21 Jul 2022

PGPH-D-21-00455R1

The translation and validation of the Organ Transplant Symptom and Well-Being instrument in China

Dear Dr. LI,

Thank you for submitting your manuscript to PLOS Global Public Health. After careful consideration, we feel that it has merit but does not fully meet PLOS Global Public Health’s publication criteria as it currently stands. Therefore, we invite you to submit a revised version of the manuscript that addresses the points raised during the review process.

We look forward to receiving your revised manuscript.

Kind regards,

Javier H Eslava-Schmalbach, M.D., Ph.D., MSc

Academic Editor

Journal Requirements:

1. Please add a full list of legends for all your Supporting Information files after the references list.

Additional Editor Comments (if provided):

Dear Authors:

Please answer or comment each one of the reviewers' suggestion in the response to reviewers' letter. Changes included should be mentioned in this letter. All the reviewers' question should be answered in the letter and highlighted within the text of the manuscript. Your did not answer the 1st reviewer comments from the 2nd (partially) to the 5th questions. Also, comments from the Editor were not answered either.

This is the most important part of the editorial process, because it is the moment when improvements to the manuscript could be done.
---

## [Editor Report · Decision Letter 2]

28 Jul 2022

The translation and validation of the Organ Transplant Symptom and Well-Being instrument in China

PGPH-D-21-00455R2

Dear mrs LI,

We are pleased to inform you that your manuscript 'The translation and validation of the Organ Transplant Symptom and Well-Being instrument in China' has been provisionally accepted for publication in PLOS Global Public Health.

Best regards,

Javier H Eslava-Schmalbach, M.D., Ph.D., MSc

Academic Editor